# Psychometric properties of the Chinese version of the Resilience Scale (RS-14): Preliminary results

Wei Chen[1,2]*, Enhui Xie[3], Xue Tian[1,2], Guyin Zhang[1,2]

1 School of Psychology, Guizhou Normal University, Guiyang, China, 2 Center for Big Data Research in Psychology, Guizhou Normal University, Guiyang, China, 3 The School of Psychology and Cognitive Science, East China Normal University, Shanghai, China

* chenweihb@yeah.net

**Data Availability Statement:** All relevant data are within the paper and its Supporting Information files.

**Funding:** This research was funded by Ministry of education humanities and social sciences research

## Abstract

### Background

In recent years, resilience has received extensive attention in psychology. The 14-Item Resilience Scale (RS-14) has been developed as a newer and shorter version of the resilience scale and has been applied in Western countries. In Eastern cultures, however, and particularly among Chinese populations, its factor structure remains unverified. The purpose of this study is to realize the first evaluation of the psychometric characteristics of the Chinese version of the RS-14 in young adults from Mainland China.

### Methods

The resilience scale, Connor-Davidson resilience scale, general health questionnaire 12, perceived stress scale 14, general self-efficacy scale and meaning in life questionnaire were used to investigate 1010 undergraduates (321 male college students, 689 female college students, aged 17–25 years; mean age = 20.27; SD = 1.572). We evaluated the item quality, latent structure, reliability, criterion validity and differential item functioning on the gender variable.

### Results

Through the analysis methods of exploratory and confirmatory factor-analytic, the original single-factor model has been proven to be applicable within the Chinese population. Both an adequate construct validity and an excellent degree of reliability were reflected in the data. In addition, test-retest evinced good stability. The current study interrogates associations with external criteria, as well as providing evidence in support of the RS-14.

### Conclusion

To sum up, this study showed that the RS-14 is a reliable assessment for measuring resilience in China, and provides an alternative to the original scale.

youth fund project (Grant no. 18XJC190001), Periodical achievements in the planning of philosophy and social sciences in Guizhou province (Grant no. 18GZYB54), Research results of humanities and social sciences project of higher education department of Guizhou province (Grant no. 2018ssd25) and Doctor project of Guizhou Normal University (Grant no. 2017).

**Competing interests:** The authors have declared that no competing interests exist.

## Introduction

In recent years, resilience, as a new concept, has become something of a hot topic [1]. According to Wagnild and Young [2], resilience is a positive personality trait that promotes individual adaptation, and in most studies, resilience refers to positive coping and adaptation in the face of loss, difficulty or adversity [3]. When changes in life pose a threat to people, this biological instinct of self-protection will be displayed, and protective factors from the individual, family, and society will interact to form a dynamic system, which will jointly resist the unfavourable influence of the environment. Resilience not only combines the latest research results of evolutionary psychology and health psychology, but also launches new discussions—as a new concept from the perspective of positive psychology [4].

Characteristics, processes and results are the three principal orientations that provide a definition of resilience [5, 6]. On the one hand, resilience is regarded as the interaction of psychological characteristics during stress [7]. On the other hand, resilience has been considered a function or behavioural outcome [8], while fewer commentators consider it to be a two-stage process allowing people to adapt [9, 10]. In the current study, following Fogarty and Perera, resilience was conceptualized as a distinguishing and very stable personality characteristic [11].

Adolescents and children are the primary focus in terms of measuring resilience. In addition, the research on resilience has produced high levels of effectiveness and reliability in communities, primary health care and psychiatric populations [12]. It may be concluded, based on substantial empirical and theoretical research, that culture exerts a direct or indirect influence on resilience, in terms of both its quality and its degree [13]. Moreover, the difference in the impact of potential religious and cultural influences on resilience—for instance, Western Christian culture and Eastern Buddhist culture—may be particularly significant. Among schizophrenic patients, their levels of resilience, which are clearly relevant to their prospects of rehabilitation, have manifested cross-cultural discrepancies [13]. Since resilience is significant in so many areas, the phenomenon merits evaluation via a dedicated, objective instrument and index.

### The Resilience Scale and its short version

Regarding the assessment of resilience, a number of measures have been developed to evaluate assumed resilience factors: more specifically, most of these tools are either based on a feature-oriented approach [14] or focus on measuring the availability of resources and protective factors to maintain or restore mental health, despite significant adverse factors [15]. Moreover, the Resilience Scale (RS) was first developed by Wagnild and Young [2] to determine the degree of individual psychological resilience. It was a 7-Likert point scale ranging from 1 (disagree) to 7 (agree). Since it comprised 25 items, the original RS came to be designated as RS-25. After principal component analysis, it was found that the structures of three factor, four factor and five factor are ambiguous, and the two-factor structure of the RS-25 (personal competence and acceptance of self and life) was seen as more suitable. The internal consistency was great ($\alpha$ = .91, $p$ < 0.001) [2]. Additionally, the RS-25 has been proven to be suitable for different environments and has undergone a series of revisions [16–18].

There is a significant correlation between depression, self-esteem, life satisfaction and various other factors (on the one hand) and resilience, as statistical evidence reveals [19, 20]. Although the RS-25 is very popular, the element structure of the RS-25 is still controversial, and some studies have demonstrated that a single-factor structure is better than the original two-factor version. In addition, the single factor of the RS-25 was generally supported by several studies and was well verified [21, 22]. Ruiz-Párraga et al. conducted an EFA within a

sample of 300 Spanish chronic-musculoskeletal-pain patients, and found that the single-factor solution fit the data best [22].

Wagnild developed the RS-14, which is a 14-item form of the RS [23]. It is derived from the original 25-item RS, but entails the removal of all items with an inter-item correlation below .40. Likewise, factor analysis of principal components method was used to support the single-factor model and demonstrated an excellent reliability ($\alpha$ = .96, p < 0.001). In order to determine the true effectiveness of this approach from a psychological perspective, in evaluating individual resilience in young adults and adolescents, additional research is clearly required [19, 24]. Kwon and Kwon conducted factor analysis of all the items of RS-14 [25]. This showed that the total variance of the two factors is 55.43%. In the research of Tian [26], the two-factor structure appeared more appropriate for detecting low resilience in cancer patients. In fact, the unidimensional structure of RS-14 is the optimal one, despite some research evidence indicating that two factors are appropriate. By using CFA, researchers also found that the unidimensional structure yielded a good fit for their data, e.g., $\chi2$ (53) = 148.297, goodness-of-fit index (GFI) = .93, comparative fit index (CFI) = .092, and root mean square residual (RMSR) = .08 [27].

More recently, through a series of EFA and CFA of the Polish version in three different samples, Surzykiewicz et al. found that the RS-14 in early adulthood samples showed a stable factor structure (e.g., RMSEA = .015, TLI = .99, CFI = .99, GFI = .99) [28]. Furthermore, the indexes in the sample of adolescents were RMSEA = .049, TLI = .94, CFI = .98, GFI = .98. The problem group showed RMSEA = .042, TLI = .93, CFI = .98, GFI = .97. Moreover, RMSEA = 0.063, GFI = 0.96, CFI = 0.94 were found by Nishi et al. [29]. According to Abiola and Udofia, the internal consistencies and concurrent validity of the RS and the RS-14 in a Nigerian sample were .87 and .81 [16]. In a sample of 430 nursing and university psychology students from Japan, the internal consistencies and test–retest reliability coefficients of the RS-14 were good [e.g., $\alpha$s were .88 and .84 for total scores, respectively; 30]. Based on a study designed to assess the Finnish version of the RS-14, Losoi et al., found the internal consistency reliability was high for the total scale ($\alpha$ > .85) [31]. In addition, RS-14 is widely used in Brazil, Italy, Greece and other countries [32–34].

Compared with the RS-25 and other scales that assess resilience, the RS-14 provides details of the pattern and profile of resilience by utilizing a widely available measure of resilience. This in turn facilitates comparison with past and future studies. In contrast to other instruments used to measure resilience, potential structure and clarity appear unstable in the context of the RS-14. Since the RS-14 is a relatively recent development, however, and particularly since it has been sparsely deployed in China, its revisions to date have been minor and limited.

## Resilience in Chinese populations

According to previous research, when studying trait resilience, it is necessary to be sensitive to social and cultural factors, which are relevant, inter alia, to how different groups are defined [28, 35]. In China, for example, the academic workload for students is extremely high, and this is particularly true for adolescents. Therefore, the psychological measurement characteristics of the RS-14 were tested in 60 adolescents from a Beijing school for students with poor behaviour. This indicated that enhanced resilience is related to life skills, while ameliorating behavioural and emotional problems is connected to "mindfulness" [36]. Chung et al. using data from 1,816 Hong Kong Chinese adolescents, concluded that it is highly important to formulate intervention measures to enhance the resilience of teenagers and promote their positive mental health [37].

Tian examined the psychometric properties of the RS-14 in two samples (625 eligible residents for the first study and 970 cancer patients for the second) [26]. Nonetheless, Tian (ibid.) maintains that the implications of RS-14 for young people and teenagers remain insufficiently explored, and this is especially notable since these are the main application groups for the scale. In addition, the RS-14 is widely used in China to screen patients with tumours, cancer and other major diseases. It assists clinicians in the timely identification of less resilient patients. Besides this, it also facilitates psychological care and psychological intervention for patients in clinical practice [38, 39]. Other research focuses on the role of resilience and interrogates its influencing factors, which is a useful approach, among other things, towards improving the psychological health of the elderly. Therefore, an attempt to provide the basis for intervention to relieve psychological pressure and promote psychological health from a new perspective is necessary [40].

Nonetheless, no study has so far adequately explored the psychometric attributes of the RS-14 among Chinese college populations, and the latter may manifest different resilience levels. The university period is a critical period for college students in the transition to adulthood. At this time, individuals are undergoing great changes in cognition and emotion. They have to face all kinds of pressures, frustrations or even traumas in life. They are more sensitive than children and adults. As a coping mechanism for college students' psychological stress, resilience can help such students develop and emerge "normally" from these negative life events. It will aid them as they struggle with frustration, trauma and stress [41]. In terms of the potential use of the RS-14 among different groups, we resolved to use it to test the psychometric characteristics of Chinese college students, while hoping eventually to develop a more accurate scale to evaluate resilience.

## The present study

There has been relatively little research explicitly to address the applicability of RS-14, mostly because of its recent development. Specifically, the present study is the first China-based research dedicated to collecting data from Chinese college students. The study will address RS-14 in terms of its factor structure, while simultaneously examining its reliability and validity. First, the main purpose of this study is to explore the factor structure of RS-14 with EFA and CFA. The reliability and construct validity of the 14 items measured by RS-14 were tested through a series of method analyses on RS-14. Moreover, we sought to investigate and test whether the one-factor structure proposed by Wagnild [23] can be replicated in Chinese college students through confirmatory factor analysis. More exactly, we sought to deploy the single-factor model of exploratory factor analysis to determine the structural effectiveness of the one-factor structure, in the context of the Chinese version of the scale.

Finally, this study aimed to test the validity of the RS-14 and to assess the correlation measurements of external standards (i.e. health status, perceived stress, self-efficacy and meaning in life). Prior pieces of research have shown that these standards are all associated with the degree of resilience [20, 42–44]. Therefore, we hypothesized that they would significantly correlate with the total score of the RS-14. Furthermore, the internal consistency and construct validity of the RS-14 would be analysed.

## Methods

### Ethics statement

This study was approved by the Ethics Committee of Guizhou Normal University in China (No. 20190615), and written informed consent was obtained from the participants.

## Participants and procedure

First, potential participants were screened to determine their suitability, and those deemed eligible were invited to take part. All students were tested in their classrooms during regular college hours. Names were not recorded on the scales, which were collected promptly on completion. Participant-related information was innocuous and was, in any case, kept strictly confidential. The study was conducted with 1,010 college students, who were first instructed regarding the study objectives. Convenient sampling for the whole class was conducted, 1,100 copies were distributed among five universities in Guiyang University Town, and 1,010 copies were retained after eliminating irregular and missing questionnaires. Two weeks later, one of the classes was retested and 53 valid questionnaires were collected.

The data were collected from a sample of 1,010 college students, which consisted of 306 males (30.3%), 689 females (68.2%) and 15 missing cases (1.49%) aged 17–25 years ($M$ = 20.27, $SD$ = 1.57), resident in Guiyang college town. The mean age of the male college students was 20.93 years ($SD$ = 1.70, range 17–25 years) and the mean age of the females was 19.98 years ($SD$ = 1.41, range 17–24 years). The data comprised odd and even halves, and the EFA data were drawn from a subset of the sample. The data from another subset of the sample were used for the CFA, and these data were deployed to examine the validity and reliability of the RS-14.

For the test-retest group, a class of students was invited to complete the research again two weeks later in order to verify the RS-14 score test-retest reliability, which was assessed by using testing times (time 1 and time 2) as the independent variables. The data were collected within a timespan of a few weeks. If we had continued to collect retest data from other participants, the two-week retest interval for the remaining participants might have caused confusion. The college students were collected from a sample of 53 college students (10 males, 42 females and 1 missing case), aged 17–21 years ($M$ = 19.28, $SD$ = .85). The mean age of the male college students was 19.44 years ($SD$ = .88) and the mean age of the female college students was 19.24 years ($SD$ = .85).

## 14-item Resilience Scale (RS-14)

The 14-item resilience scale [the RS-14; 45] was used to assess the degree of resilience, and was a short version of the original resilience scale [the RS; 2]. The RS-14 was a 14-item and single-factor structure instrument, comprising a 7-point Likert scale ranging from 1 (strongly disagree) to 7 (strongly agree). Examples of the items were, "I feel that I can handle many things at a time" and, "I keep interested in things". The Chinese translation of the RS-14 was developed with a backtranslation procedure to ensure accuracy. Suitable internal consistency was suggested by a figure of .917 for the coefficient alpha (regarding the RS-14 of the original scale). Further, the original RS-14 was retested after three weeks, and the test-retest reliability of RS-14 was .736 (.595, .837).

## Connor-Davidson Resilience Scale (CD-RISC)

We used the Connor-Davidson resilience scale (CD-RISC) to access measurable pressure-response capability [46]. This was done in accordance with the simplified CD-RISC study, which was scored via Likert-5 points and comprised 25 items. Specifically, the range extended from 0 points ("not true at all") to 4 points ("true nearly all the time"). The higher the resilience of the individual concerned, the greater the total score he/she should register on the scale. The Chinese version of CD-RISC by Yu and Zhang, which only extracted three potential factors, was used to distinguish the characteristics of those who performed well and those who did not perform well after adversity, and evinced good applicability in China [47]. With a value of

alpha was .880, the present study demonstrated an excellent degree of internal consistency in terms of the total scale.

### General Health Questionnaire 12 (GHQ-12)

Present mental health and psychological disturbance were measured by using the general-health questionnaire 12 [48], which contained a total of 12 items. The GHQ-12 used a Likert-4 rating scale (1 = less than usual, 2 = no more than usual, 3 = rather more than usual, 4 = much more than usual). The GHQ-12 gives a total score of 36 or 12 points according to the selected scoring method. High scores equate to poor mental health—the higher the score, the poorer the health. For the current study, the GHQ-12 demonstrated high validity and reliability, making it a useful tool to assess general health; alpha was .693.

### Perceived Stress Scale 14 (PSS-14)

The personal evaluation and perception of life situations reflect the degree of stress in college students, which was assessed via the perceived stress scale [49]. PSS-14 is a 14-item self-report scale with seven positive stated items and seven reverse-coding items. Each item is rated on a Likert-5 point scale: 0 = never, 1 = almost never, 2 = once in a while, 3 = often, 4 = very often. The higher the score, the higher the perceived pressure level; the lower the score, the lower the perceived pressure level. The question, "In the past month, how often have you been upset because of something that happened unexpectedly" was presented to participants. Coefficient alpha was .784 in this study.

### General Self-Efficacy Scale (GSES)

The general sense of perceived self-efficacy was measured with the general self-efficacy scale (GSES) of Schwarzer and Jerusalem [50]. The perception and adaptive behaviour of individuals confronted with challenges were conceptualized via this scale, which comprised 10 items. The GSES used a Likert-4 rating scale, specifically: 1 = not at all true, 2 = hardly true, 3 = moderately true, 4 = exactly true, indicating a neutral position on an item. The range was from 10 to 40 points and an example of the items was, "I can remain calm when facing difficulties because I can rely on my coping abilities". Moreover, the GSES in its Chinese version demonstrated good predictive capacity and was shown to be unidimensional [51]. In the present study, the coefficient alpha was .626.

### Meaning in Life Questionnaire (MLQ)

The meaning in life questionnaire was used to access the participants' concrete values, as well as "value experiences" that the individuals encountered or espoused within their lives [52]. The scale comprised 10 items, rated on a Likert-7 point scale ranging from 1 (absolutely untrue) to 7 (absolutely true). The scale included five items related to the "search for meaning" and a further five to ascertain "the presence of meaning". The scores pertaining to the two sense subscales were calculated separately. Specifically, two subscales indicated the presence of meaning (MLQ-P) and the search for meaning (MLQ-S). In the current study, the Chinese version of the MLQ was used [53] and internal consistency was .816 for the total scale.

### Data analysis

All analyses used the total score and each discrete item score. First, calculations were made to determine descriptive statistics for the separate RS-14 items. Second, in order to determine the factor structure, we conducted a factor analysis on the odd and even numbers, to split the data

as an exploratory factor analysis (EFA) using STATA MP 13.1 [54]. The other half of the sample was deployed to locate and identify the optimal model for data explanation. Mplus 7.4 was used for the confirmatory factor analysis (CFA) [55].

Firstly, the present study provided preliminary descriptive statistics, including means, standard deviations, skewness and kurtosis, which are given in Table 1. Skewness and kurtosis are the numerical characteristics of random variables in probability theory. Skewness is a quantity that reflects the distribution shape of random variables. If skewness is less than 0, then kurtosis is biased to a smaller numerical value than that of standard normal distribution. Positive skewness indicates that the peak is biased to a larger value than the standard normal distribution [56, 57]. Kurtosis, meanwhile, indicates the thickness of the distribution tail and reflects the shape in which random variables are distributed. In order to test the normal distribution of samples, skewness and kurtosis, which have proven to be powerful and informative tests, need to be explained [56, 58]. If kurtosis is > 0, it means that its distribution is relatively sharp compared with the standard normal distribution. Conversely, if kurtosis is negative, this means that its distribution is relatively flat compared with the standard normal distribution [56, 57].

Next, the RS-14 was developed through a series of EFA and CFA, which were effective methods of presenting construct validity with few assumptions (the research instrument measured the degree of expected structure) [59]. The scree plot is one of the common methods used to establish the number of factors within EFA; the eigenvalue must be greater than 1 [60, 61]. The main purpose of EFA is to determine the number of factors of the RS-14, together with the degree of correlation between each factor and each observation variable. Parallel analysis (PA) is a method frequently used to establish consensus regarding how many components should be retained. It is a novel and less rigorous methodology within EFA [62, 63].

Following Hayton et al. [64], by comparing the scree plot of eigenvalues in real data with the curves of the average eigenvalues of a group of random matrices, the intersection point of the two eigenvalue curves is found, and the absolute maximum number of factors to be extracted is determined according to the position of the intersection point. If the eigenvalues of the real data fall on the average eigenvalue curve of the random matrix, these factors are retained. Otherwise, these factors are discarded. Henson and Roberts recommended the simultaneous use by researchers of various criteria, including parallel analysis, carefully to determine the quantity of factors to be retained [65]. Therefore, the minimum average partial correlation

**Table 1. Each item—Descriptive statistics.**

| Item | M | SD | Skewness | Kurtosis |
|------|------|-------|----------|----------|
| Item1 | 4.46 | 1.455 | -.560 | -.174 |
| Item2 | 4.68 | 1.234 | -.462 | .037 |
| Item3 | 5.11 | 1.271 | -.920 | .693 |
| Item4 | 5.36 | 1.309 | -.924 | .621 |
| Item5 | 5.08 | 1.489 | -.768 | .148 |
| Item6 | 5.13 | 1.330 | -.713 | .189 |
| Item7 | 4.66 | 1.331 | -.433 | -.153 |
| Item8 | 4.98 | 1.316 | -.575 | .039 |
| Item9 | 4.89 | 1.299 | -.604 | .040 |
| Item10 | 4.88 | 1.445 | -.639 | -.147 |
| Item11 | 5.14 | 1.298 | -.682 | .207 |
| Item12 | 4.80 | 1.356 | -.522 | -.117 |
| Item13 | 5.07 | 1.367 | -.697 | .142 |
| Item14 | 5.42 | 1.401 | -.987 | .739 |

[MAPC; 66] was also used. In the case of k components, this extracts 0 ~ (k-1) principal components in a gradually increasing manner, and when the partial correlation of the mean square root reaches the minimum, the factor extraction stops. Generally speaking, PA and MAPC both demonstrated parsimony and completeness [67].

Further, CFA aims to examine the ability of models with predefined factors to fit actual data and if data obeys the normal distribution maximum likelihood (ML) which is a common method of estimation within CFA [68] will be used. In terms of the RS-14 and its related scales, the consistency of its structure with single-factor structures was investigated via factor analytic research. TLI and CFI have an excellent fitting degree of 0.95, and the standard can be relaxed appropriately. Moreover, TLI and CFI were 0.90, indicating an adequate fit [69, 70]. Based on the recommendations of Taasoobshirazi and Wang [71], the following criteria were sufficient to indicate the goodness-of-fit of the model to the data: RMSEA = .06, Standardized Root Mean Square Residual (SRMR) = .08.

The study should, moreover, determine the degree of measurement error, i.e. the reliability [72]. Furthermore, we sought to verify the correlation between our scale as a whole, items of the RS-14, and external variables. Reliability is expressed by the correlation coefficient; the α result is a number between 0 and 1. Specifically, a correlation coefficient αs < .70 shows a deficiency; .70 to .89 indicates an acceptable or even good reliability, and a value greater than .90 suggests excellent reliability [73]. The higher the correlation, the higher the internal consistency.

The third step involved in the assessment of differential item functioning (DIF) is to test whether the item response has measurement invariance between different groups. DIF—which comprised the two forms of uniform and non-uniform DIF—addressed individuals with similar psychological characteristics. The fairness of the test relates to the context of questionnaire validity. Therefore, based on the ordered Logistic regression model, this study applied a likelihood ratio to test the significance of the regression coefficient (the formula follows below) and to analyse DIF [74, 75]. Finally, in order to determine whether the significant difference was influenced by the inherent DIF deviation of some items, the independent sample t-test was deployed.

$$f(resonse|traitlevel, group) = \beta_0 + \beta_1 \times traitlevel + \beta_2 \times group$$
$$+ \beta_3 \times (group \times traitlevel)$$

In the above formula, $f$ on the left represents an ordered Log function, instead of a general linear function.

Finally, the total scores of the RS-14 and other measures were used to examine criterion validity. This is another instrument for measuring the same variable, and correlation can be used to determine the ways in which different instruments measure the same variable [72]. Criterion validity was assessed by determining the degree of Pearson correlation between the total score of the RS-14 and the external criteria. Following Terwee et al., the coefficients exceeding approximately r = .40 were considered satisfactory and indicated that the RS-14 was an effective evaluation measure [76].

## Results

### Descriptive statistics

The elementary descriptive statistics of the 14 items were indicated by factor analysis. All the skewness and kurtosis statistics were within the acceptable range (skewness within ±3.00 and

**Table 2. Item-total and corrected item-total correlations, critical ratio, commonality and factor loading of the RS-14.**

| Item | ITC | CITC | CR | Commonality | Factor Loading |
|---|---|---|---|---|---|
| Item1 | .662 | .593 | 22.030*** | .495 | .637 |
| Item2 | .756 | .711 | 28.736*** | .652 | .748 |
| Item3 | .694 | .639 | 22.365*** | .512 | .679 |
| Item4 | .660 | .598 | 19.674*** | .512 | .633 |
| Item5 | .574 | .490 | 18.203*** | .328 | .516 |
| Item6 | .728 | .675 | 25.649*** | .526 | .701 |
| Item7 | .646 | .581 | 23.717*** | .436 | .612 |
| Item8 | .747 | .698 | 28.627*** | .564 | .729 |
| Item9 | .732 | .680 | 28.005*** | .577 | .709 |
| Item10 | .670 | .602 | 23.956*** | .468 | .639 |
| Item11 | .752 | .704 | 26.081*** | .574 | .723 |
| Item12 | .736 | .683 | 28.936*** | .534 | .711 |
| Item13 | .666 | .602 | 23.293*** | .447 | .619 |
| Item14 | .732 | .676 | 26.248*** | .534 | .708 |

*Note*. ITC = item-total correlation; CITC = corrected item-total correlation; CR = critical ratio;

***$p < .001$.

kurtosis within ±8.00 which indicating normally distributed [57, 58]. Table 1 provides a detailed breakdown of these data.

## Item analysis

The item analysis of the RS-14 was conducted using the following methods (see Table 2). The first method involves identifying the product-moment correlation coefficient between the individual items and the total RS-14 score. The correlation coefficient between each item and the total score was between .577 and .756, while the corrected correlation coefficient was between .494 and .712. The second method finds the critical ratio (CR), with 27% of the total score of the RS-14 items as the boundary point, and divides it into high groups and low groups. Then, as we use an independent sample *t*-test, if each item can achieve a significant level, this indicates that the item discrimination is high. For the 14 items, the results demonstrated a t-value much higher than 3. Further, CRs of the RS-14 were between 18.427 and 29.099, reaching an extremely significant level ($p < .001$). One may thus assume a higher degree of item discrimination for the RS-14. The third method is to perform a factor analysis based on all data. The commonality of the 14 items was between .328 and .652, while the factor loading was between .516 and .748.

## Construct validity

Based on the serial numbers of the paper questionnaires using EpiData 3.1, 1010 valid data were divided in two via odd and even numbers. Half the data were deployed for CFA, and the other half for EFA. EFA was carried out on the odd-numbered data, indicating that EFA could be conducted effectively (KMO = 0.779, Bartlett's test = $\chi^2 / df \approx 3214.522/91 \approx 35.324\ p < .001$). The results showed that there was only one eigenvalue (6.746) above 1 generated by the factor analysis of principal components method, which accounted for 48.18% of the total variance. Factor analysis evinces certain obvious shortcomings, e.g. strong subjectivity, discrepancies in the eigenvalues and the scree plot for the number of the selecting factors, etc. Therefore, PA [62, 63] and MAPC [66] were used in the present study (see Table 3). The results showed

**Table 3. The result of parallel analysis and minimum average partial correlation.**

| Factor Number | Factor Loading (Eigenvalue) | Minimum Average Partial Correlation |
|---|---|---|
| 0 | - - - - - - | .198 |
| 1 | 6.897 | .015 |
| 2 | .918 | .021 |
| 3 | .824 | .027 |
| 4 | .737 | .037 |
| 5 | .659 | .051 |
| 6 | .586 | .069 |
| 7 | .557 | .088 |
| 8 | .513 | .117 |
| 9 | .458 | .161 |
| 10 | .427 | .215 |
| 11 | .397 | .316 |
| 12 | .394 | .470 |
| 13 | .360 | 1 |
| 14 | .273 | |
| Recommended standards | >1 | Minimum |

that traditional factor analysis can choose two common factors, while PA and MAPC correlation analysis only support the result of one common factor. Therefore, only one factor is recommended.

CFA was applied to the even-numbered data, and the adequacy of the single-factor model was demonstrated by the fit indices: $\chi^2 / df \approx 301.503/77 \approx 3.916$, $p < .001$, CFI = .929, TLI = .916, RMSEA = .078 (070, .088) [The 95% Bootstrap confidence interval of percentile is in parentheses, and the number of Bootstrap is 2000. If there is no special explanation, the same below], SRMR = .040.

## Reliability

The coefficient alpha of the overall RS-14 is .917 (.907, .925). After a two-week interval, the Pearson correlation coefficient of the revised first instance and the second instance of the RS-14 was .736 (.595, .837).

## Differential Item Functioning (DIF)

With a view to assessing the significance of the regression coefficient of the Logistic regression model, via the application of a likelihood ratio, the present study utilized DIF [77, 78]. The existence of non-uniform DIF for four items was suggested by the fact that, for items 2, 3, 4 and 7 and for both male and female students, the likelihood ratio tests yielded significant results. (See Table 4).

## Criterion validity

Following research on the reliability and validity of the RS-14 [46, 47], the CD-RISC, the GHQ-12, the PSS-14, the GSES and the MLQ were selected as criteria (see Table 5).

## Discussion

For the first time in the research field, the current study aimed, while providing preliminary results in the context of a Chinese population, to examine the psychometric properties and

**Table 4. Assessment of DIF across gender groups.**

| Item | Uniform DIF | | Non-Uniform DIF | |
|---|---|---|---|---|
| | Change in Est. | Whether DIF or not | p-value | Whether DIF or not |
| Item1 | .000 | NO | 1 | NO |
| Item2 | .010 | NO | .490 | NO |
| Item3 | .000 | NO | .128 | NO |
| Item4 | .005 | NO | .875 | NO |
| Item5 | .004 | NO | .118 | NO |
| Item6 | .001 | NO | .408 | NO |
| Item7 | .003 | NO | .077 | NO |
| Item8 | .002 | NO | .978 | NO |
| Item9 | .004 | NO | .173 | NO |
| Item10 | .001 | NO | .350 | NO |
| Item11 | .004 | NO | .474 | NO |
| Item12 | .003 | NO | .967 | NO |
| Item13 | .000 | NO | .786 | NO |
| Item14 | .007 | NO | .982 | NO |

*Note*. Uniform DIF is present if the change in estimation of regression coefficient is greater than 0.1. Non-uniform DIF exists if the Bonferroni-corrected p-value for the regression coefficient for those items is less than .05.

factor structure of the RS-14 of Wagnild [23]. The item analysis of the RS-14 was duly conducted and suggested high item discrimination. In addition to the internal consistency, good reliability was evinced by the correlations among the various RS-14 items. After a series of EFA and CFA, adequate construct validity indicated that the original single-factor model of the RS-14 fit the data well. A high gender discrepancy among the young-adult participants was evinced by the DIF results. Moreover, the correlations between the total score of the RS-14 and external criteria demonstrated the validity of the test-score interpretations.

Consistently with the work of Wagnild [23], only one eigenvalue obtained from the original RS-14 model was above 1, which satisfied the standard in psychometrics for the present samples. Nevertheless, some controversy about the model structure of RS-14 still existed, as the two-factor model is more suitable for Spanish populations [79]. Very few studies have supported two factors [79], whereas the widespread use of RS-14 in recent years is closely related to its single-factor structure stability. Findings were consistent with most previous research

**Table 5. Correlations among the RS-14 and criterion scales.**

| | 1 | 2 | 3 | 4 | 5 | 6 |
|---|---|---|---|---|---|---|
| 1 RS-14 | 1 | | | | | |
| 2 CD-RISC | .519*** | 1 | | | | |
| 3 GHQ12 | -.302*** | -.231*** | 1 | | | |
| 4 PSS14 | -.415*** | .414*** | .466*** | 1 | | |
| 5 GSES | .337*** | .441*** | -.238*** | -.372*** | 1 | |
| 6 MLQ | .822** | .420*** | -.153*** | -.228*** | .231*** | 1 |

*Note*.

$^{*}p < .05,$

$^{**}p < .01,$

$^{***}p < .001.$

results, indicating that the single-factor model was strict [30, 31]. For instance, factor analysis of principal components method indicated a single-factor solution, accounting for 45.4% of the variance [33]. Pascoe, Rahman explained 48% of the variance in the whole sample as confirming a unidimensional structure [80]. Specifically, for the present study, 48.18% of the total variance for the students was the result of a single eigenvalue above 1 [32, 35].

In addition, the CFA results supported the factor structure derived from EFA. As expected, the single-factor structure of the RS-14 fit our data adequately and was characterized by good construct validity [23]. The single-factor model demonstrated stability, in line with earlier pieces of research [28]. Specifically, CFI = .929, TLI = .916, RMSEA = .078, SRMR = .040. All indexes were excellent, and consistent with prior research [26, 28]. Compared with the research regarding our sample group, the data for a total Italian sample indicated CFI = .91, RMSEA = .008, SRMR = .007 [33]. According to Aiena et al. [35], the criteria were as follows: CFI = .93, TLI = .92, RMSEA = .11, SRMR = .04. Moreover, a good level of fitness for the single-factor model (e.g., CFI = .93, TLI = .91, RMSEA = .59, SRMR = .041) was found by Damásio, Borsa [32]. In summary, the cross-cultural consistency of the RS-14 is borne out by the results of the present study and by the findings of earlier research [19, 23].

The consistency evaluation results of the initial scale (α = .917) proved the internal consistency of the RS-14 Chinese version. Similarly, adequate internal consistency (α = .88) among college students was indicated by the Japanese data [30]. The high internal consistencies were .91 [24] and .89 [34], which suggested that the RS-14 was reliable. Furthermore, test-retest was used in the present study and we found that reliability was acceptable (α = .736). This was determined 14 days after the initial evaluation, with reference to 53 young participants. The results differed from the results found in the literature and it was found that the RS-14 had good reliability. The existing documents provide evidence regarding the determination of the test-retest reliability of the instrument. This reinforces the ability of the instrument to identify variations of resilience over time. In further research, test-retest reliability was used to prove the stability of the measure over time.

Further, related instruments (e.g., general health, the personal evaluation and perceived stress) provided precise evidence for the criterion validity of the RS-14 scores. The present study was consistent with prior research, insofar as the CD-RISC was associated with the RS-14 in terms of access and resilience. Although the two instruments measure the same characteristics and show the relationship of dependency, there still exists a theoretical and functional difference. The CD-RISC was developed on the basis of coping, stress and adaptation research [15]. Although the CD-RISC was designed for use on clinical mental-health sites, the RS-14 has been used in numerous applications [81]. Moreover, it will be necessary, in future, to give full rein to the comprehensive functionality of the RS-14 and broaden its applications.

Likewise, our findings suggested that resilience was significantly negatively correlated to general health scores and perceived stress, which was consistent with previous studies [82, 83]. Specifically, the higher the resilience, the better the health condition and the more easily young people can manage pressure. Generally, resilience may be a result of exposure to extreme stress [15] and resilience can affect the perception and toleration of pressure to a certain extent, when an individual is in a "difficult situation" [84]. Conversely, young people of low resilience are ill equipped to deal with negative emotions. This has negative implications for physical and mental health, in terms of anxiety and depression [6, 85].

The present study found an alternative and specific explanation of resilience, via external indicators of general self-efficacy, and meaning in life—which was consistent with prior research [86–88]. Indeed, the results showed that higher levels of self-efficacy correlated, perhaps unsurprisingly, with greater resilience. Martínez-Martí and Ruch also identified widely

recognized resilience-related factors (i.e. positive self-efficacy, self-esteem and social support) [20].

Individuals with high self-efficacy tend to be optimistic about their future prospects. They thus tend to persevere in the face of adversity and are not easily disheartened by pressure. Resilience refers to the adaptation to difficulties or problems, correct cognition and active responses, including the understanding of meaning in our lives. The association of resilience with a coherent sense of "life meaning" is an established and longstanding one [89].

As noted, the current research provides data support for the single-factor model of the RS-14 [23]. Further, this reinforces the use of the RS-14 as a short and readily available, well-validated assessment in Chinese populations. Likewise, resilience is a common trait not only possessed by the Chinese, but also demonstrated by human beings all over the world. In other words, resilience is a cross-cultural trait. In sum, the validated RS-14 could be used among young adults in different research settings and provide evidence for the concept of resilience in early life.

## Limitations

The relationship between meaning in life and resilience is not the province of the present study. Nonetheless, the two elements are connected, and research on this connection is sparse. Within future research, it will be important to address this relationship, and this will also require a careful distinction between the two components of meaning in life—namely, the search for meaning and the presence of meaning. Improvements in measurement and understanding will facilitate the promotion of both resilience and "life meaning". Another potential limitation is that the sample age in this study is relatively homogeneous, but resilience is a concept significant throughout life. We know that the RS-14 may be applied to the elderly and to adults, as well as adolescents. In spite of the limitations reflected in the current study, however, the Chinese version of the RS-14 was shown to be a satisfactory measuring and assessment tool for young adults.

## Supporting information

**S1 Data. Data 1 for descriptive statistics, item analysis and criterion validity.** Data 2 for EFA. Data 3 for CFA. Data 4 for test-retest reliability coefficient. Data 5 for DIF by gender. (RAR)

## Acknowledgments

We are especially grateful to the participants who volunteered to participate and gave their valuable time during their learning time.

## Author Contributions

**Data curation:** Enhui Xie, Xue Tian, Guyin Zhang.

**Funding acquisition:** Wei Chen.

**Writing – original draft:** Wei Chen, Enhui Xie.

**Writing – review & editing:** Wei Chen, Guyin Zhang.

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
