## [Editor Report · Decision Letter 0]

15 Jun 2020

PONE-D-20-08907

Psychometric Properties of the Chinese Version of the Resilience Scale (RS-14): Preliminary Results

PLOS ONE

Dear Dr. chen,

Thank you for submitting your manuscript to PLOS ONE. After careful consideration, we feel that it has merit but does not fully meet PLOS ONE’s publication criteria as it currently stands. Therefore, we invite you to submit a revised version of the manuscript that addresses the points raised during the review process.

Before sending out your manuscript for external review, I ask you to specify the estimation method that was used in CFA. Furthermore, I wonder why you explained skewness and Kurtosis a lot but you reported nothing about skewness and Kurtosis in the results. Please write a sentence at least.

We look forward to receiving your revised manuscript.

Kind regards,

Gian Mauro Manzoni, Ph.D., Psy.D.

Academic Editor

PLOS ONE

2. Please ensure that you refer to Figure 1 in your text as, if accepted, production will need this reference to link the reader to the figure.

---

## [Author Response · Author response to Decision Letter 0]

7 Jul 2020

Thank you, editor.

“Before sending out your manuscript for external review, I ask you to specify the estimation method that was used in CFA.” 

Response: We added the estimation method in paper. According to the skewness and kurtosis, the maximum likelihood (ML)estimation was selected in CFA. 

Furthermore, I wonder why you explained skewness and Kurtosis a lot but you reported nothing about skewness and Kurtosis in the results. Please write a sentence at least.

Response: We added the related instructions in the results. 

“All the skewness and kurtosis statistics were within the acceptable range (skewness within ±3.00 and kurtosis within ±8.00 which indicating normally distributed” .

---

## [Decision Letter · Decision Letter 1]

19 Oct 2020

Psychometric Properties of the Chinese Version of the Resilience Scale (RS-14): Preliminary Results

PONE-D-20-08907R1

Dear Dr. chen,

We’re pleased to inform you that your manuscript has been judged scientifically suitable for publication and will be formally accepted for publication once it meets all outstanding technical requirements.

Kind regards,

César Leal-Costa, Ph. D

Academic Editor

PLOS ONE

Additional Editor Comments (optional):

Reviewers' comments:

Reviewer's Responses to Questions

**Comments to the Author**

1. If the authors have adequately addressed your comments raised in a previous round of review and you feel that this manuscript is now acceptable for publication, you may indicate that here to bypass the “Comments to the Author” section, enter your conflict of interest statement in the “Confidential to Editor” section, and submit your "Accept" recommendation.

Reviewer #1: All comments have been addressed

Reviewer #2: All comments have been addressed

2. Is the manuscript technically sound, and do the data support the conclusions?

Reviewer #1: Yes

Reviewer #2: Yes

3. Has the statistical analysis been performed appropriately and rigorously? 

Reviewer #1: Yes

Reviewer #2: Yes

4. Have the authors made all data underlying the findings in their manuscript fully available?

Reviewer #1: Yes

Reviewer #2: Yes

5. Is the manuscript presented in an intelligible fashion and written in standard English?

Reviewer #1: Yes

Reviewer #2: Yes

6. Review Comments to the Author

Reviewer #1: The present study succeeds to develop a new scale to assess resilience in Chinese. The manuscript seems OK to me. No other comments are added.

Reviewer #2: The paper has been improved with the revision made by the authors. It is now a publishable article. Good work.

7. PLOS authors have the option to publish the peer review history of their article (what does this mean?). If published, this will include your full peer review and any attached files.

Reviewer #1: No

Reviewer #2: No

---

## [Editor Report · Acceptance letter]

22 Oct 2020

PONE-D-20-08907R1 

Psychometric Properties of the Chinese Version of the Resilience Scale (RS-14): Preliminary Results 

Dear Dr. chen:

I'm pleased to inform you that your manuscript has been deemed suitable for publication in PLOS ONE. Congratulations! Your manuscript is now with our production department. 

Kind regards, 

on behalf of

Dr. César Leal-Costa 

Academic Editor

PLOS ONE